

# Systematic analysis of JmjC gene family and stress-response expression of KDM5 subfamily genes in *Brassica napus*

Xinghui He[1,2,3], Qianwen Wang[3], Jiao Pan[1,2,3], Boyu Liu[1,2,3], Ying Ruan[1,2,3] and Yong Huang[1,2,3,*]

[1] Key Laboratory of Crop Epigenetic Regulation and Development, Hunan Province, Changsha, China
[2] Key Laboratory of Plant Genetics and Molecular Biology of Education Department, Changsha, Hunan Province, China
[3] College of Bioscience and Biotechnology, Hunan Agricultural University, Changsha, Hunan Province, China
[*] These authors contributed equally to this work.

Corresponding author
Yong Huang,
yonghuang@hunau.edu.cn,
hycncs@163.com

## ABSTRACT

**Background**. Jumonji C (JmjC) proteins exert critical roles in plant development and stress response through the removal of lysine methylation from histones. *Brassica napus,* which originated from spontaneous hybridization by *Brassica rapa* and *Brassica oleracea*, is the most important oilseed crop after soybean. In JmjC proteins of *Brassica* species, the structure and function and its relationship with the parents and model plant *Arabidopsis thaliana* remain uncharacterized. Systematic identification and analysis for JmjC family in *Brassica* crops can facilitate the future functional characterization and oilseed crops improvement.

**Methods**. Basing on the conserved JmjC domain, JmjC homologs from the three *Brassica* species, *B. rapa* (AA), *B. oleracea* (CC) and *B. napus,* were identified from the *Brassica* database. Some methods, such as phylogenic analysis, chromosomal mapping, HMMER searching, gene structure display and Logos analysis, were used to characterize relationships of the JmjC homologs. Synonymous and nonsynonymous nucleotide substitutions were used to infer the information of gene duplication among homologs. Then, the expression levels of *BnKDM5* subfamily genes were checked under abiotic stress by qRT-PCR.

**Results**. Sixty-five JmjC genes were identified from *B. napus* genome, 29 from *B. rapa,* and 23 from *B. oleracea*. These genes were grouped into seven clades based on the phylogenetic analysis, and their catalytic activities of demethylation were predicted. The average retention rate of *B. napus JmjC* genes (*B. napus JmjC* gene from *B. rapa* (93.1%) and *B. oleracea* (82.6%)) exceeded whole genome level. JmjC sequences demonstrated high conservation in domain origination, chromosomal location, intron/exon number and catalytic sites. The gene duplication events were confirmed among the homologs. Many of the *BrKDM5* subfamily genes showed higher expression under drought and NaCl treatments, but only a few genes were involved in high temperature stress.

**Conclusions**. This study provides the first genome-wide characterization of JmjC genes in *Brassica* species. The *BnJmjC* exhibits higher conservation during the formation process of allotetraploid than the average retention rates of the whole *B. napus* genome. Furthermore, expression profiles of many genes indicated that *BnKDM5* subfamily genes are involved in stress response to salt, drought and high temperature.

## INTRODUCTION

Epigenetics refers to heritable change for gene function that occurs without a change in DNA sequence and can dynamically regulate global gene expression through reversible chemical modifications on DNA and histones in eukaryotic chromatin (*He & Cole, 2015*). Epigenetic regulation mainly includes acetylation, phosphorylation, histone methylation, DNA methylation, and small non-coding RNAs. Histone modification is an important epigenetics mechanism. Various post-translational covalent modifications, which primarily occur on histone (H3, H4, H2A, and H2B) lysines and arginines residues, form "histone code" to regulate various biological processes (*Bannister & Kouzarides, 2011*). Histone methylation is usually catalyzed by three protein families of histone methyltransferases: protein arginine methyltransferase family, Su (var)3-9, Enhancer-of-zeste and Trithorax (SET) domain family, and telomeric silencing disruptor that is also known as DOT1-Like (Kmt4/DOT1L) (*Greer & Shi, 2012*). Histone lysine methylation, playing many different roles in biological processes ranging from heterochromatin formation to transcription regulation, is dynamically regulated by histone lysine methyltransferases (KMTs) and histone lysine demethylases (KDMs), and can be distinguished depending on the position of lysine residue and the number of added methyl groups in lysine residues, which carry mono-, di-, or tri-methylated groups (*Liu et al., 2010a*; *Liu et al., 2010b*).

Histone modifications can influence gene expression to regulate the plant response to stress, including cold, freezing, saline, drought and submergence (*Bej & Basak, 2017*). Genome-wide H3K4 methylation patterns (H3K4me1, H3K4me2 and H3K4me3) show dynamic responses to dehydration stress in *Arabidopsis thaliana* (*Van Dijk et al., 2010*). *Arabidopsis* ATX1, H3K4me3, is involved in dehydration response through ABA-dependent and -independent pathways (*Ding, Avramova & Fromm, 2011*). Under drought stress, H3K4me3 enrichment is correlated with the activation of *Arabidopsis* drought stress-responsive genes, such as *RD29A* and *RD20* (*Kim et al., 2012*; *Qiao & Fan, 2011*). H3K4me3 can be maintained at low levels after rehydration and which could function as an epigenetic mark of drought stress memory (*Kim et al., 2012*). Under heat stress, H3K4 methylation accumulates to activate gene expression and can be sustained after heat stress to positively respond on a future stress incident (*Lämke et al., 2016*). Rice genome-wide H3K4me3 profiling showed positively correlation with the transcript level of drought stress-responsive genes (*Zong et al., 2013*). Histone methylation is also a reversible process regulated by methyltransferases and three distinct classes of demethylases. KDMs mainly consist of LSD1/KDM1s (Lysine specific demethylase 1) and JmjC-domain enzymes, which both utilize oxidative mechanisms. JmjC, a highly conserved domain, was first reported by Takeuchi and colleagues in 1995 and was named as JmjC domain in 2000 (*Balciunas & Ronne, 2000*; *Takeuchi et al., 1995*). This domain carries eight $\beta$-sheets forming enzymatically-active pocket with three conserved and

necessary amino-acid residues for binding with Fe (II) cofactor and two additional residues for binding with $\alpha$-ketoglutarate ($\alpha$KG) (*Chen et al., 2006*; *Klose, Kallin & Zhang, 2006*). *Arabidopsis* JmjC proteins are divided into five subfamilies: KDM4/JHDM3 (AtJMJ11-13), KDM5/JARID1 (AtJMJ14-19), JMJD6 (AtJMJ21/22), KDM3/JHDM2 (AtJMJ24-29) and JmjC domain-only (AtJMJ20 and AtJMJ30-32) (*Luo et al., 2013*). The H3K4 methylases and demethylases dynamically balance the H3K4 methylation status among H3K4me1, H3K4me2 and H3K4me3, to maintain the optimum level of H3K4 methylation and adapt to external environment. KDM5 is a specific subfamily that specifically removes H3K4 methylation modifications. However, most reports on H3K4 demethylase functions were mainly focused on regulating plant development. For example, AtJMJ14/PKDM7B, a histone H3K4 demethylase, represses floral integrators *Flowering Locus T* (*FT*), *AP1*, *SOC1* and *LFY* during vegetative growth (*Lu et al., 2010*; *Yang et al., 2010*). AtJMJ15 regulates flowering time by demethylating H3K4me3 at *Flowering Locus C* (*FLC*) chromatin (*Yang et al., 2012b*). AtJMJ18 is dominantly expressed in companion cells exhibiting H3K4me3 and H3K4me2 demethylase activity of *FLC*. *atjmj18* mutation results in a weak late-flowering phenotype, and its overexpression induces early-flowering (*Yang et al., 2012a*). Moreover, the overexpression of *AtJMJ15* may regulate gene expression that enhances stress tolerance (*Shen et al., 2014*). Although several functions of H3K4 methylation modifications in response to abiotic stresses have been reported, only a few were evaluated. *Brassica* species might have diverged from a common ancestor with an *Arabidopsis* lineage from 14.5–20.4 million years ago (*Yang et al., 1999*). Allotetraploid species *Brassica napus* (oilseed rape, AACC, 2n = 38) originated from interspecific spontaneous hybridization between *Brassica rapa* (AA, 2n = 20) and *Brassica oleracea* (CC, 2n = 18) (*Yang et al., 2010*). The protein organization and function of JmjC domain in *Brassica* species and its relative relationship with model plant *Arabidopsis* remain uncharacterized. *B. napus* is currently the most important oilseed crop, preceded only by soybean. However, *B. napus* is vulnerable to abiotic stress that limits its growth and productivity and reduces its economic benefits. KDM5/JARID1 subfamily may regulate many abiotic stress responses genes through down-regulated H3K4me3 and H3K4me2 but the roles of H3K4 demethylation in abiotic stress remain unknown.

## MATERIALS AND METHODS

### Identification of Jmjc Proteins and Chromosomal Map Construction

The JmjC protein sequences of *Arabidopsis thaliana* (AtJMJ11-22 and AtJMJ24-32) were obtained as our previously described by Huang et al. in 2016 (File S1), and these sequences were used as queries to BLASTp JmjC proteins of *B. rapa*, *B. oleracea* and *B. napus* in the *Brassica* database (http://brassicadb.org/brad/index.php/blastPage.php). The JmjC protein sequences in *Oryza sativa* were retrieved from Phytozome database (Version 12). The result sequences of BLASTp were confirmed using both SMART (HMMER) and NCBI (BLASTp) with default parameters, and proteins without JmjC domain were excluded. The loci information of JmjC gene was used to generate chromosome maps by the Mapchart, and the retention rates were calculated based on homologous genes on corresponding chromosome (*Voorrips, 2002*).

## Analysis of JmjC sequences

The gene structures were visualized by GSDS (http://gsds.cbi.pku.edu.cn/). DOG program was used to sketch site information of domain organization (*Ren et al., 2009*). Multiple sequence alignment which is based on the full-length protein sequences is performed by ClustalW (*Thompson, Higgins & Gibson, 1994*), and its resulting files were subjected to phylogenic analysis by neighbor-joining method in MEGA7.0 program with pairwise deletion, p-distance model and Bootstrap test of 1000 replicates (*Tamura et al., 2013*). The multiple sequence alignment result were subjected to phylogenic analysis by Maximum likelihood with pairwise deletion, Nearest-Neighbor-Interchange and Bootstrap test of 1000 replicates. Proteins sequences of JmjC were aligned with ClustalW to create Logo maps (http://weblogo.berkeley.edu/logo.cgi), and the Fe(II) binding sites are showed in red triangle and $\alpha$KG binding site are showed in black triangle.

## Duplicated JmjC Genes in *B. napus*

The duplicated genes of *B. napus* were defined by the method of *Yang et al. (2008)* and *Sun et al. (2015)*. The CDS sequence coverage and amino acid identity were determined by Blastn/Blastp. The number of non-synonymous mutations (Ka) and the number of synonymous substitutions (Ks) of duplicated genes were calculated by DnaSP 6.0 (*Town et al., 2006*). The duplication time was calculated by $T = Ks/2 \lambda \times 10^{-6}$ ($\lambda = 1.5 \times 10^{-8}$) (*Koch, Haubold & Mitchell-Olds, 2000*).

## Plant material and stress treatment

The *Brassica napus* L. ssp Xiangyou 15 was used as the plant material of stress treatment. The seeds were provided by Key Laboratory of Crop Epigenetic Regulation and Development in Hunan Province, Hunan Agricultural University. *B. napus* seedlings were grown on clay substrates at 22 °C chamber in a 16 h light/8 h dark photoperiod. One-month old plants with 4 true leaves were treated. For drought stress, the seedlings were grown without watering, and leaves were sampled at 0, 5, 10, and 15 days. For salt stress, seedlings were treated with 0, 100, 200, and 300 mM NaCl, and leaves were harvested at 3 days after treatment. For high temperature stress, seedlings were grown at 40 °C, and leaves were harvested at 0, 12, 24, and 36 h after treatment. All harvested samples were immediately frozen in liquid nitrogen. Three independent biological replicates for each treatment were conducted.

## RNA Extraction and Real-Time Quantitative PCR (RT-qPCR)

Samples RNA was extracted by TRIzol reagent kit (Invitrogen, Carlsbad, CA, US) and reverse transcribed into cDNA by Revert Aid RT Kit (Thermo Fisher, USA). The specific primer pairs (File S4) used for real-time PCR with Fast Start Universal SYBR Green Master (ROX) (Roche, Switzerland) on a CF x 96 Real Time System (BIORAD). The *BnActin* gene (accession ID: NC_027768) was used as reference gene. Each sample was run in triplicate and their expression levels were analyzed by $2^{-\Delta\Delta}$ method (*Livak & Schmittgen, 2001*).

## RESULTS

### Chromosome maps of JmjC genes in *Brassica*

In this study, 21 *Arabidopsis* JmjC proteins were used as queries to Blastp in *Brassica* genomics (http://brassicadb.org/brad/index.php/blastPage.php). *B. napus* carried 65 JmjC genes, whereas its parents *B. rapa* and *B. oleracea* had 29 and 23, respectively (File S1). 57 JmjC genes of *B. napus* were mapped on 19 chromosomes (AACC, $2n = 38$), and 8 genes were still on scaffolds, in which 6 were from Cn subgenome and 2 from An subgenome. In addition, 23 *B. oleracea* genes and 29 *B. rapa* genes lactated on C01-C09 and A01-A10 chromosomes, respectively.

The JmjC genes in An and Cn subgenomes of *B. napus* show nearly identical distributions to its ancestor genomes *B. rapa* (A-genome, 29) and *B. oleracea* (C-genome, 23) (Fig. 1). A02 and A07 chromosomes only exist in one member of *B. napus*, which is similar to its ancestor *B. rapa* genomes. A09 chromosome carries the highest number, seven genes. Four tandem JmjC genes pairs located on chromosomes A03, A09, and C03 in *B. napus* (Fig. 1). The tandem duplicated genes *BnJMJ27;e* and *BnJMJ27;f* on A03 subgenome are derived from *BrJMJ27;a* and *BrJMJ27;b*, which belong to *AtJMJ27* orthology. Tandem duplicated gene pairs *BnJMJ27;d/BnJMJ27;b* and *BnJMJ17;a/BnJMJ17;c* might have resulted from the forming processes allotetraploidy of *B. napus*. However, *BnJMJ27;a* and *BnJMJ27;g* of C03 subgenome are absent from the ancestor *B. oleracea* genomes, and the orthologous genes of these tandemly duplicated genes appear in the corresponding location of A03 subgenome, which indicate that *BnJMJ27;a* and *BnJMJ27;g* might have derived from the cross duplication of A03 subgenome. *BnJMJ31;a* of C03 subgenome, *BnJMJ18;c* of C08 subgenome, *BnJMJ17;b,* and *BnJMJ29;c* of C09 subgenome may have similar origins.

### Phylogenetic analysis of JmjC proteins in *B. napus*

A phylogenetic tree was constructed with 21 JmjC proteins form *Arabidopsis*, 19 from *O. sativa*, 29 from *B. rapa*, 23 from *B. oleracea*, and 65 from *B. napus* to examine their relationships. To make the name of JmjC gene more coherent and rational, JmjC genes of *Brassica* were named based on their relationship to homologous gene in *Arabidopsis*. The NJ tree that has similar topology with ML tree and more high bootstrap values, is used to analyze the phylogenetic relationship of JmjC proteins (Fig. 2; File S3). The JmjC proteins of *Brassica* were divided into seven clades, except BoJMJ19;c and BoJMJ19;d: KDM4/JHDM3, KDM5A and B, JmjC domain only A and B, JMJD6 and KDM3/JHDM2 groups. This classification pattern was similar to the one previously reported for JmjC-domain proteins in the green lineage (*Huang et al., 2016*). JmjC, Jumonji N (JmjN) and zinc-finger (ZnF) motifs were the special motifs for KDM4/JHDM3; JmjC, JmjN, F/Y-rich N terminus (FYRN) and F/Y-rich C terminus (FYRC) for KDM5A; JmjC, JmjN and plant homeodomain (PHD) for KDM5B; JmjC and F-box for JMJD6; JmjC and RING (really interesting new gene) for KDM3/JHDM2; and JmjC domain for JmjC domain only A/B (Figs. 3–7). JmjN domain specifically exists in all proteins of KDM4/JHDM3, KDM5A and KDM5B group, except BrJMJ14;a, BnJMJ16;b, BrJMJ17 and BnJMJ17;c. *B. rapa*, *Arabidopsis* and *O. sativa* possess similar amounts of JmjC proteins in KDM5B, JmjC domain-only A, JMJD6 and JmjC domain only B group. However, *B. oleracea* does not have

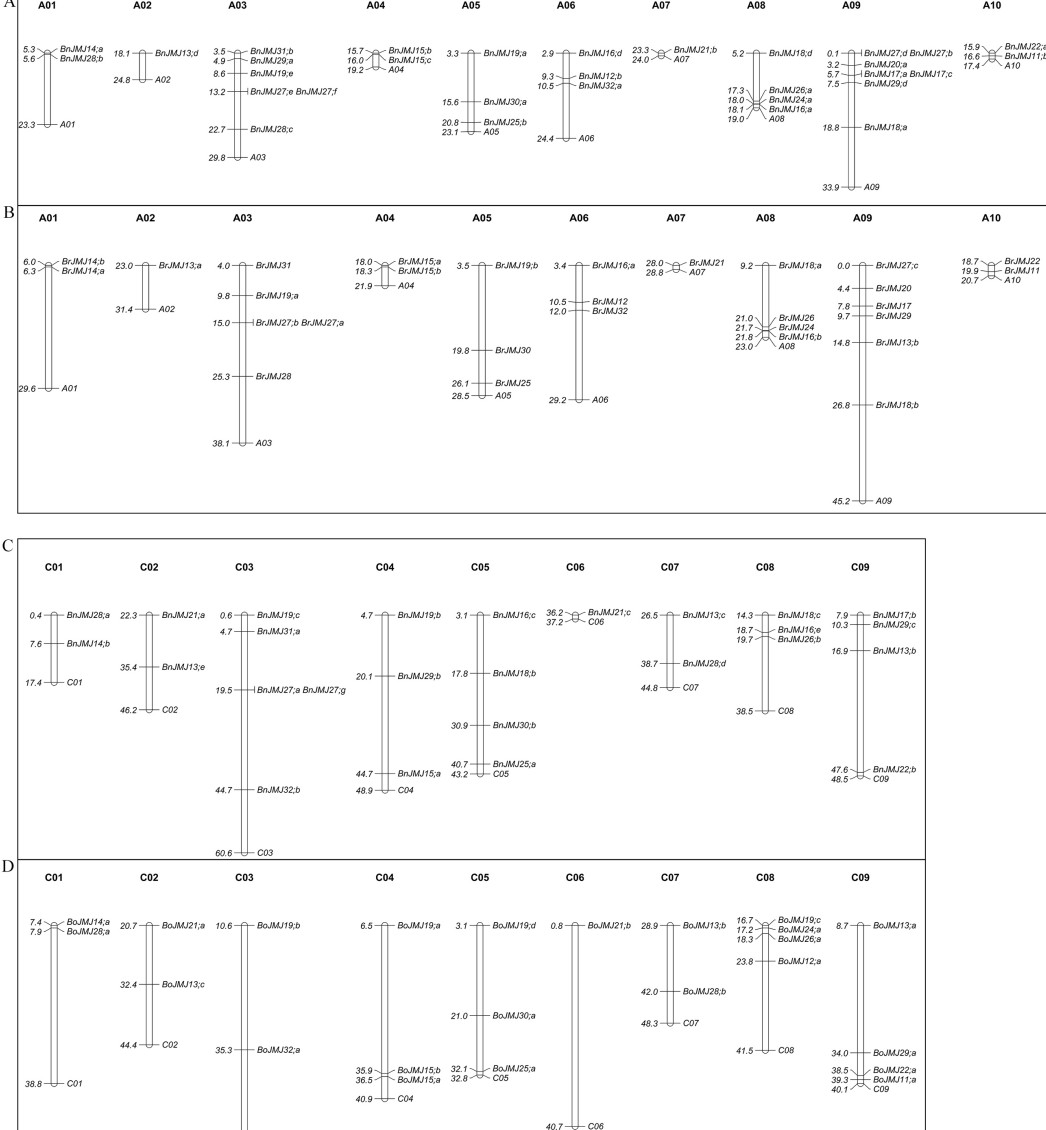

**Figure 1** **Chromosomal distribution of *Brassica* genes.** *Brassica* genes (57 *B. napus*, 23 *B. oleracea* and 29 *B. rapa*) was mapped on chromosomes except eight scaffolds genes of *B. napus*: (A) *B. napus* genes distribution of A-genomics, (B) *B. rapa* genes distribution, (C) *B. napus* genes distribution of C-genomics, (D) *B. oleracea* genes distribution. The scale on the chromosome represents megabases (Mb).

JmjC protein of KDM5B and JmjC domain-only A. *B. napus* has 63 JmjC proteins, which is more than the sum of those for *B. oleracea* and *B. rapa* (Fig. 2; File S1). The gene pairs imply the closest relatives within the phylogenetic tree. JmjC phylogenetic tree identified 39 sister pairs consisting of 22 An-Ar and 17 Cn-Co (Fig. 2). Moreover, most of the sister pairs are also paralogous gene pairs between the An and Cn subgenomes (Fig. 2).

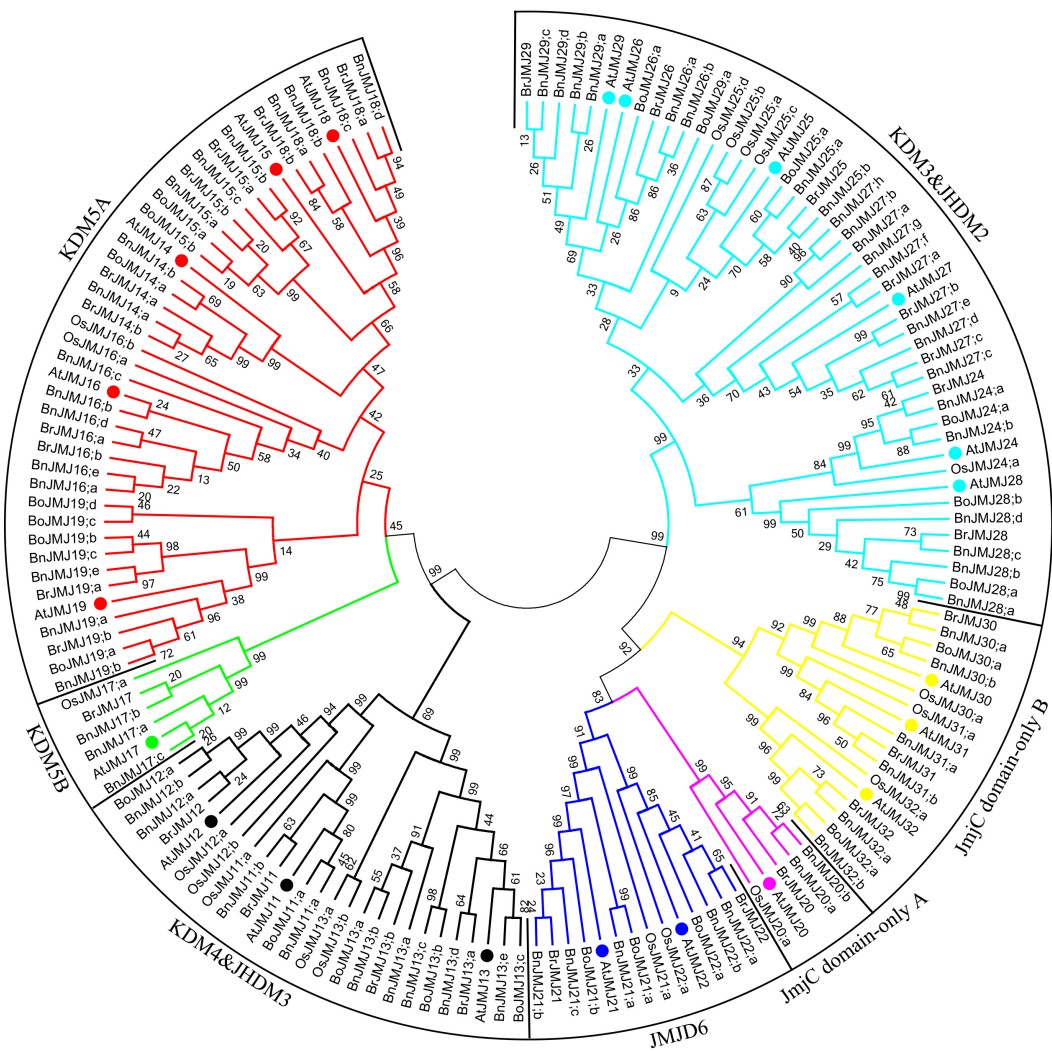

**Figure 2** **Phylogenetic tree of JmjC domain proteins.** The Phylogenetic tree included 21 JmjC domain-containing proteins form *Arabidopsis thaliana*, 19 from *Oryza sativa*, 29 from *Brassica rapa*, 23 from *Brassica oleracea* and 65 from *Brassica napus*. The JmjC domain proteins can be grouped into seven groups based on the phylogenetic tree and domain organization. Different colors show different groups. JmjC domain protein sequences were aligned using ClustalW, and the phylogenetic tree analysis was performed using MEGA7.0.The trees were constructed with the following settings: tree inference as neighbor-joining; include sites as pairwise deletion option for total sequences analysis; substitution model as p-distance.

## Group KDM4/JHDM3

Group-KDM4/JHDM3 contains nine JmjC proteins from *B. napus*, four from *B. rapa*, five from *B. oleracea*, five from *O. sativa* and three from *Arabidopsis* (Fig. 3). Group-KDM4/JHDM3 can be divided into two subgroups according to phylogenetic relationship, domain characteristic and gene structure: subgroup-I with eight *Brassica* members and two *Arabidopsis* homologous genes, *AtJMJ11* and *AtJMJ12*.

The domain organization of subgroup-I members show highly-conserved and shared JmjC, JmjN and ZnF domain. Subgroup-II contains 10 *Brassica* members and *Arabidopsis*

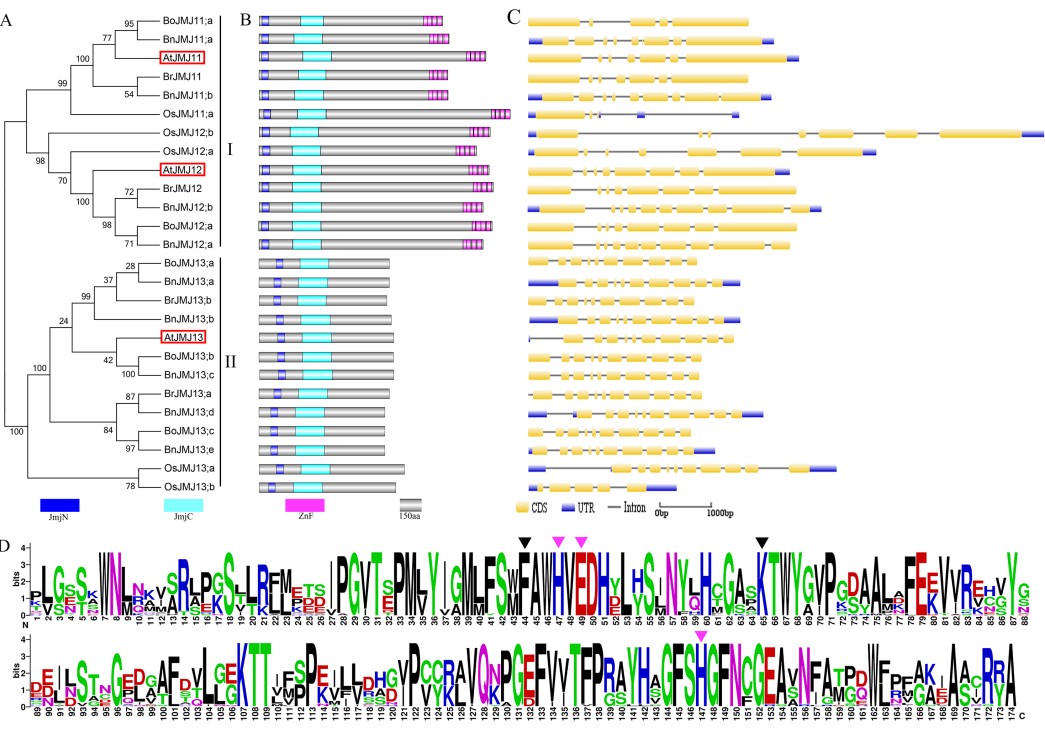

**Figure 3** **The schematic diagrams of Group-KDM4/JHDM3.** (A) Phylogeny tree, (B) domain organization, (C) gene structure, (D) logos analysis of JmjC domain.

homologous gene *AtJMJ13* shared JmjC, and JmjN (Figs. 3A and 3B). KDM4 subfamily shares the JmjN and JmjC motifs. JmjN domain is the second highly-conserved domain that is close to the N terminus and shorter than JmjC domain (*Balciunas & Ronne, 2000*). The four tandem array ZnF domain of RELATIVE OF EARLY FLOWERING 6 (REF6)/AtJMJ12 targets motif CTCTGYTY, and the ZnF domain only exists in subgroup-I (*Li et al., 2016*). REF6 also tends to bind to hypo-methylated CTCTGYTY motifs in vivo (*Qiu et al., 2019*). Subgroup-I generally harbors 7–8 exons, but subgroup-II keeps highly similar gene structures with 10 exons (Fig. 3C).

JmjC proteins have been discovered as Fe(II)- and αKG-dependent histone demethylases (*Chen et al., 2006*; *Klose, Kallin & Zhang, 2006*). The JmjN and JmjC domains, two non-adjacent domains, interact with each other through two "z-sheets and form a single functional unit to ensure the stability and appropriate transcription activity of Gis1 and maintain the overall protein levels and function of Jhd2 H3K4-specific demethylase in budding yeast (*Huang et al., 2010*; *Quan, Oliver & Zhang, 2011*). KDM4/JHDM3 has conserved Fe(II) binding site (His and Glu residues) and αKG binding site (Phe and Lys residues) (Fig. 3D).

## Group-KDM5A/B

KDM5/JARID1 further can be divided into two groups: KDM5A and KDM5B (Fig. 2; File S2). Group-KDM5A contains 18 JmjC proteins from *B. napus*, 2 from *O. sativa*, 10 from *B.*

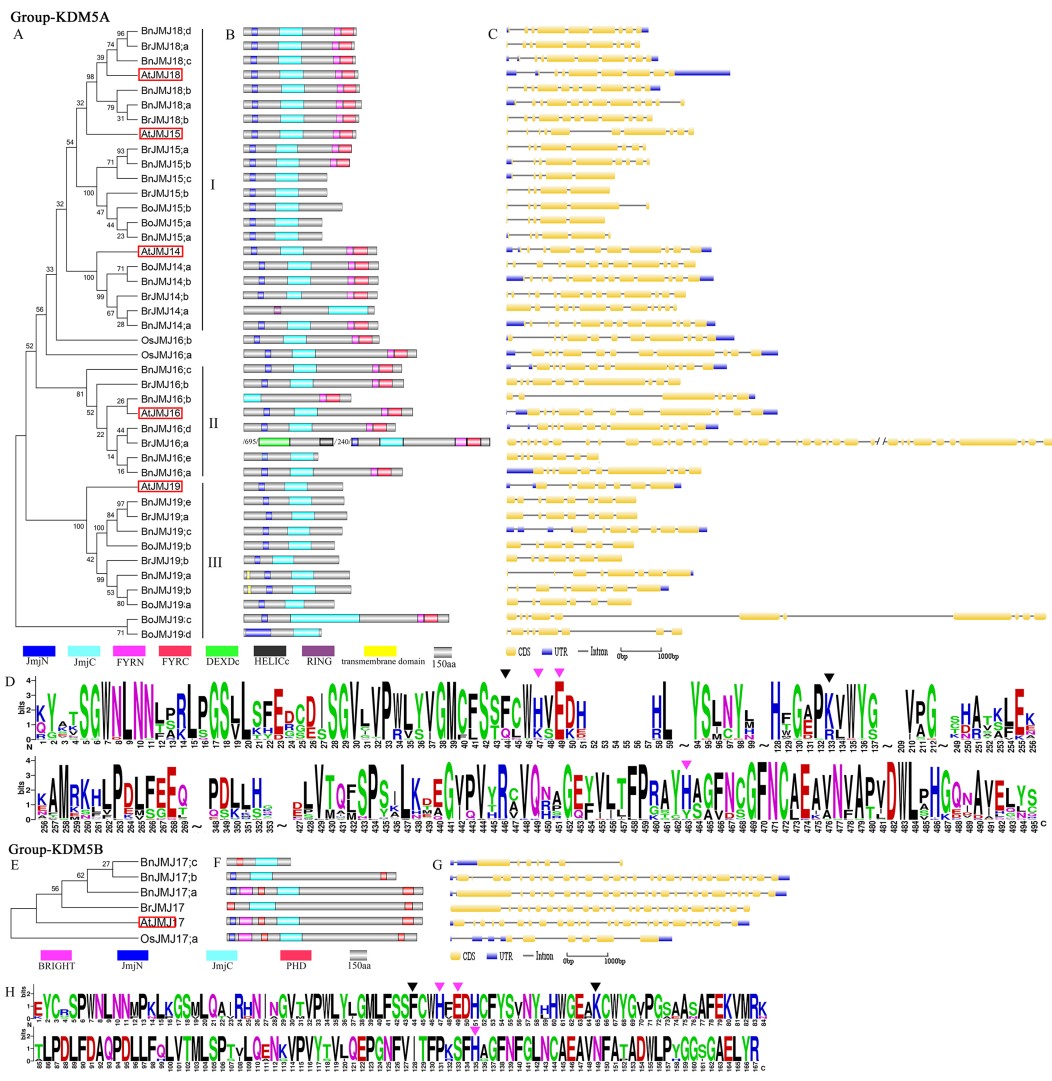

**Figure 4** **The schematic diagrams of Group-KDM5A/B.** (A & E) Phylogeny tree, (B & F) domain organization, (C & G) gene structure, (D & H) logos analysis of JmjC domain.

*rapa*, 7 from *B. oleracea* and 5 from *Arabidopsis* (Fig. 4A). Group-KDM5B only has 4 JmjC proteins from *Brassica* and 1 from *Arabidopsis* homologous gene *AtJMJ17*. *B. oleracea* does not have KDM5B JmjC proteins, and *B. napus* carries 3 members.

Group-KDM5A is distinguished by JmjC, JmjN, FYRN and FYRC motifs (Fig. 4B), and can be further divided into three subgroups: subgroup-I with 18 *Brassica* members and 3 *Arabidopsis* homologous genes, *AtJMJ14*, *AtJMJ15* and *AtJMJ18*. These members show highly-conserved domain organization sharing JmjC, JmjN, FYRN, and FYRC domains, except *BrJMJ14;a*. The phylogenetic tree showed that *B. oleracea* does not have *AtJMJ18* homologues. Moreover, *BrJMJ18;b* is clustered with *BnJMJ18;a* and *BnJMJ18;b*, as well as *BrJMJ18;a* with *BnJMJ18;c* and *BnJMJ18;d* (Fig. 4A). However, *B. napus* does not have a gene clustered with *BrJMJ14;b*, *BoJMJ15;b*, and *BrJMJ15;b*. Subgroup-II has seven *Brassica*

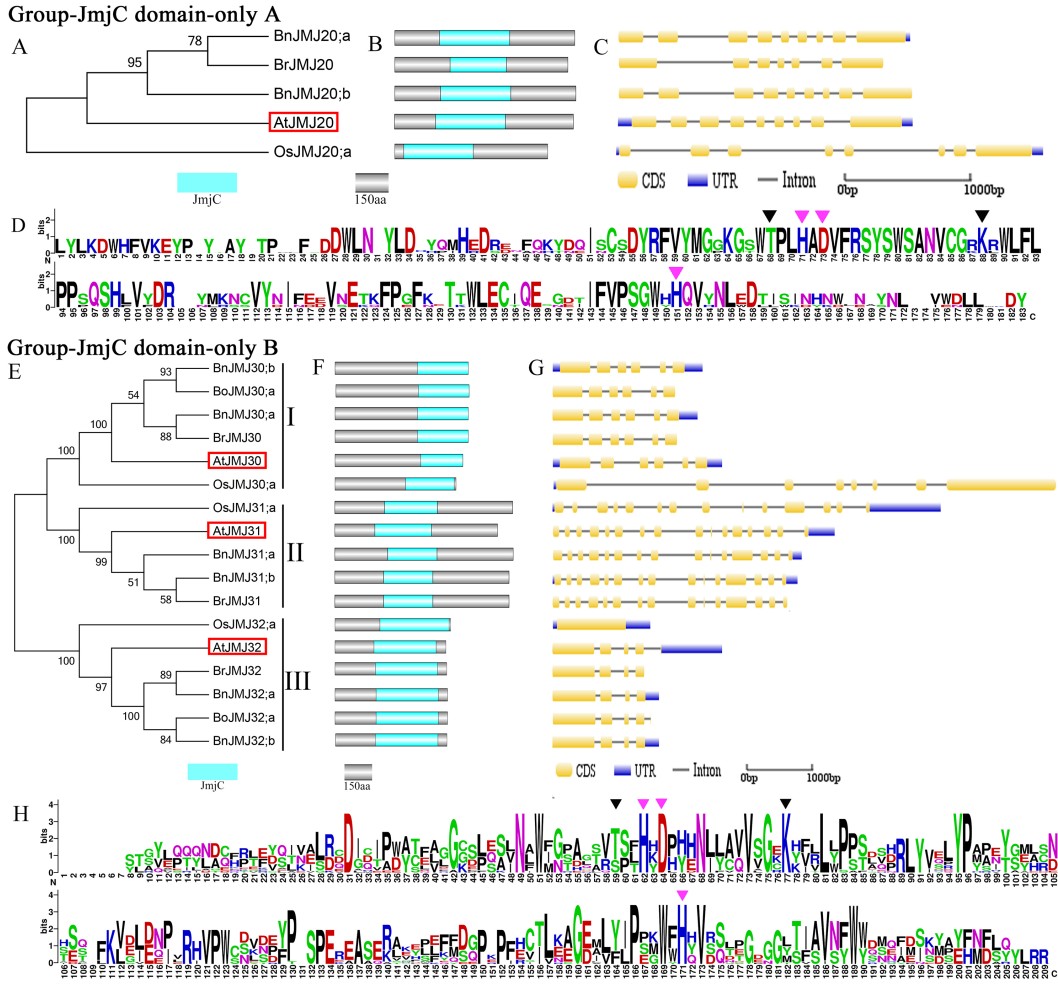

**Figure 5** **The schematic diagrams of Group-JmjC domain-onlyA/B.** (A & E) Phylogeny tree, (B & F) domain organization, (C & G) Gene structure, (D & H) Logos analysis of JmjC domain.

members and *Arabidopsis* homologous gene *AtJMJ16* sharing JmjC, FYRN, JmjN and FYRC domains, except *BnJMJ16;e,* which display highly-conserved domain organization, in addition to *BrJMJ16;a* with additional helicase superfamily C-terminal and DEAD-like helicases superfamily domains. Subgroup-III has 10 *Brassica* members and *Arabidopsis* homologous genes *AtJMJ19* and share JmjC and JmjN domains, besides BnJMJ19;a and BnJMJ19;b with an additional transmembrane domain. This finding suggested that subgroup-III may have a relatively stable inheritance during the formation process of allotetraploidy.

Group-KDM5B differs from group-KDM5A group in domain organization, which has BRIGHT and PHD but lacking FYRN and FYRC (Fig. 4F). BRIGHT is associated with H3K4 demethylase by DNA binding motif (CCGCCC) to regulate transcription (*Tu et al., 2008*). PHD mainly exerts epigenetic effectors capable of recognizing or "reading" post-translational histone modifications and unmodified histone tails (*Musselman &*

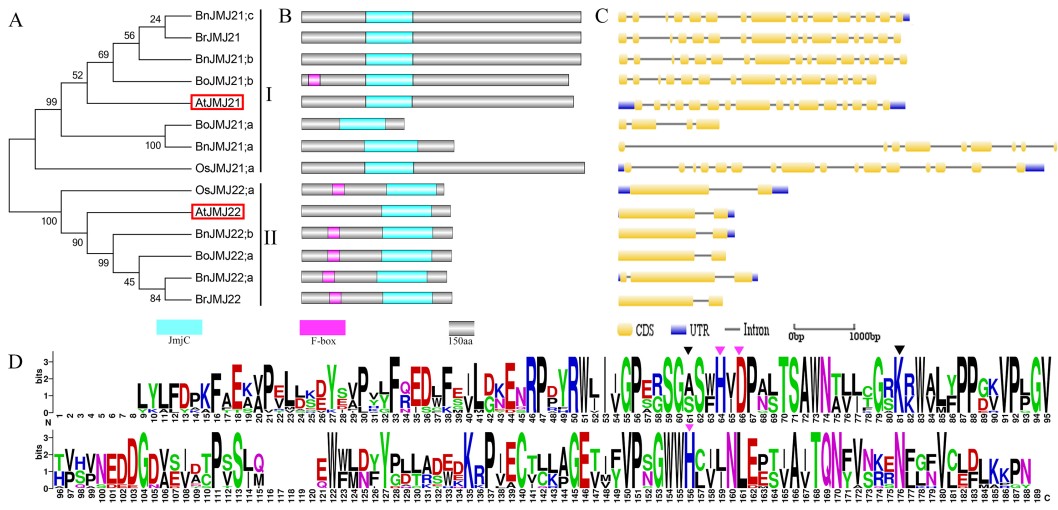

**Figure 6** **The schematic diagrams of Group-JMJD6.** (A) Phylogeny tree, (B) domain organization, (C) gene structure, (D) logos analysis of JmjC domain.

*Kutateladze, 2011*). The original PHD role in gene transcription is acted as a reader of H3K4me3 in 2006 (*Wysocka et al., 2006*). Many sophisticated functions of PHD were also determined, including H3K9me3 recognition and binding to the N-terminus of H3, indicating its key roles in regulating transcription and chromatin structure (*Wang et al., 2015*). All members of group-KDM5B group have BRIGHT or PHD domains (Fig. 4F), indicating their involvement in demethylation using JmjC domain associated with BRIGHT and PHD domains.

Group-KDM5A/B shows a wide range intron/exon number (5–36), but sister gene pairs are relatively conserved in gene structure (Fig. 4). In group-KDM5A, subgroups-I/II are highly conserved in Fe(II) and αKG binding sites, except BrJMJ16;b in which Phe is replaced by Met in αKG binding site, and His is replaced by Arg Fe(II) binding site. In subgroup-III, Phe is replaced by Gln in αKG binding site, and BoJMJ19;c/d is variable in other Fe(II) and αKG binding sites (Fig. 4D). In group-KDM5B, *BnJMJ17a* gene structure is similar to its parent *BrJMJ17* (Fig. 4G). Group-KDM5B is highly conserved in Fe(II) and αKG binding sites, similar to KDM4/JHDM3 group (Fig. 4H).

## Group-JmjC Domain-only A/B

Group-JmjC domain-only A/B and JMJD6 are distributed in different branches of a large clade. Group-JmjC domain-only A is close to group-JMJD6 but far from group-JmjC domain-only B. Group-JmjC domain-only A and B have same domain organization and only exist in JmjC domain (Fig. 2).

Group-JmjC domain-only A possesses the least number of JmjC proteins among the groups (Figs. 2 and 5A) and contains three *Brassica* members and one *Arabidopsis* homologous gene *AtJMJ20*. *B. oleraace* is lack of Group-JmjC domain-only A JmjC proteins (Fig. 5). *BnJMJ20;b* shares coincident gene structures, domain organizations and chromosomal map with *BrJMJ20* (Fig. 1, 5B and 5C) indicating that the former may have
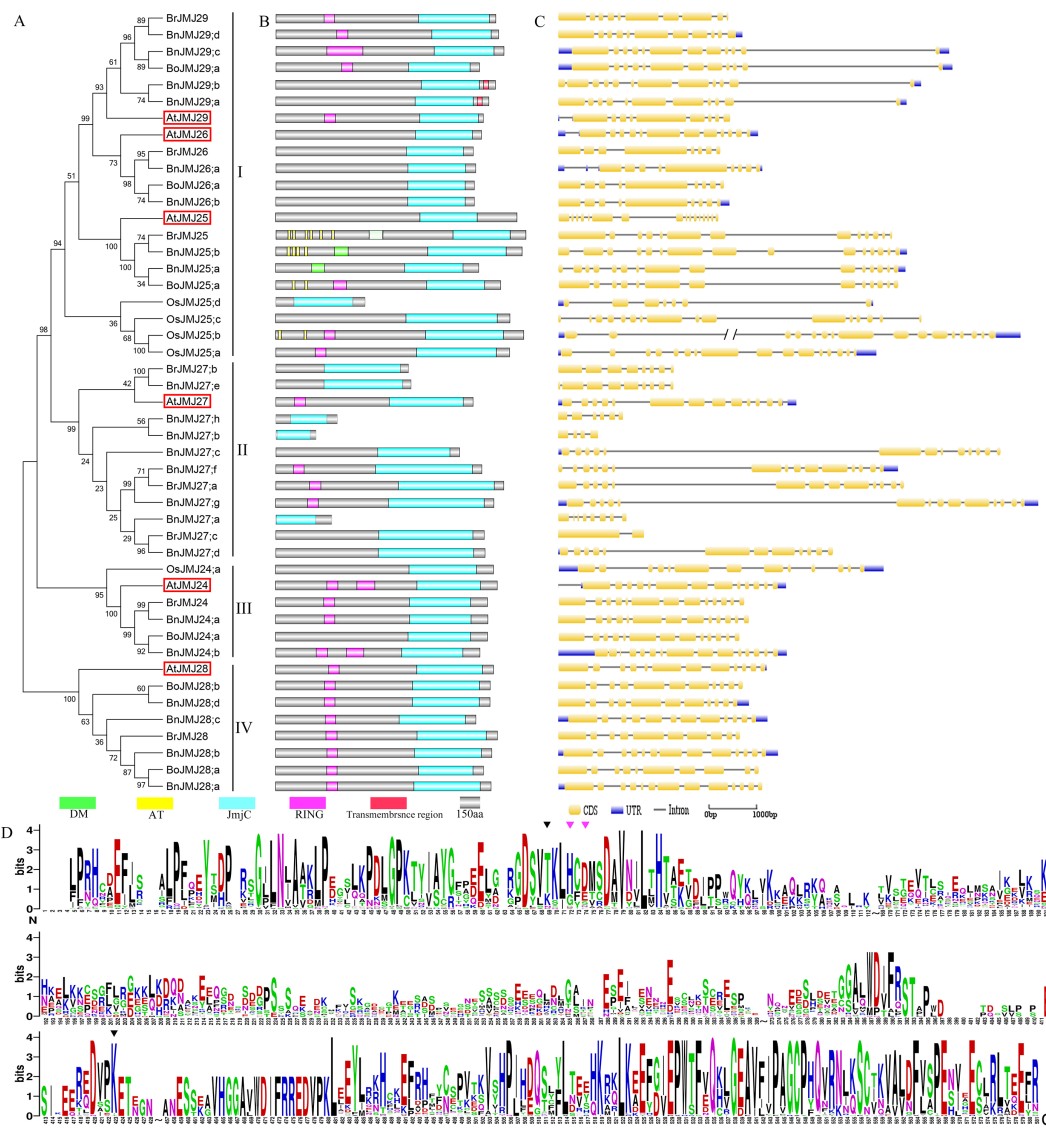

**Figure 7 The schematic diagrams of Group- KDM3&JHDM2.** (A) Phylogeny tree, (B) domain organization, (C) gene structure, (D) logos analysis of JmjC domain.

originated from the latter. Chromosomal map, CDS cover and protein ID reveal that BnJMJ20;a might be the duplicate of BnJMJ20;b (File S2).

JmjC domain-only B contains 17 JmjC proteins: 6 from *B. napus*, 2 from *B. oleracea*, 3 from *B. rapa*, 1 from *O. sativa* and 3 from *Arabidopsis*. Group-JmjC domain-only B can be further divided into three subgroups. Subgroup-I contains four *Brassica* members and *Arabidopsis* homologous gene *AtJMJ30* (Fig. 5E). Subgroup-II contains three *Brassica* members and *Arabidopsis* homologous gene *AtJMJ31*. Subgroup-III contains four *Brassica* members and *Arabidopsis* homologous gene *AtJMJ32* (Fig. 5E). Subgroups-I and III show high conservation during the forming process of allotetraploid. *B. napus* perfect inherited JmjC genes from its parents *B. oleracea and B.rapa*: BnJMJ30;b originating from

BoJMJ30;a and BnJMJ30;a from BrJMJ30;a within subgroup-I; BnJMJ32;b originating from BoJMJ32;a and BnJMJ32;a from BrJMJ32 in subgroup-III. *B. oleracea* lacks JmjC proteins in subgroup-II. BnJMJ31;a exhibits notable similarity with BnJMJ31;b in terms of domain component and gene structure, indicating that BnJMJ31;a may have originated from the inserted duplicate of BnJMJ31;b belonged to paralogues gene (Figs. 5F and 5G).

Group-JmjC domain-only A has stable exon distribution harboring approximately 7–9 exons. Sequence alignment and logos analysis of JmjC domain reveal that JmjC domain-only A group is highly conserved in Fe(II) and $\alpha$KG binding sites. However, compared with that in the KDM4/JHDM3 group, Phe is replaced by Thr in Fe(II) binding site (Fig. 5D). In group-JmjC domain-only B, subgroup-I genes contains 6 exons, subgroup- III harbors 4 exons, and subgroup- II has many exons (Fig. 5G). As compared with that in the KDM4/JHDM3 group, the Phe residue is replaced by Ser within AtJMJ31 orthology (Fig. 5H).

### Group-JMJD6

The phylogenetic tree showed that the JMJD6 group is close to JmjC domain-only A group and includes five JmjC proteins from *B. napus*, three from *B. oleracea*, two from *Arabidopsis,* two from *B. rapa* and two from *O. sativa*. Each JmjC gene of *B. napus* is clustered with a corresponding homologous gene from *B. oleracea* or *B. rapa* (Figs. 1 and 6A).

On the basis of phylogenetic tree analysis and schematic diagrams, group-JMJD6 can be further divided into two subgroups (Fig. 6). Subgroup-I contains six *Brassica* members and *Arabidopsis* homologous gene *AtJMJ21* having only JmjC domain, besides BoJMJ21;b protein with an additional F-box domain (Figs. 6A and 6B). Subgroup-II contains four *Brassica* members and AtJMJ22, which shares JmjC and F-box domains except AtJMJ22 missing F-box domain. However, their gene structure shows high conservation (Fig. 6). F-box domain recognizes a wide array of substrates and regulates many important biological processes by degrading cellular proteins in plants (*Gupta et al., 2015*).

Subgroup-I generally harbors 15–16 exons, except BoJMJ21;a (4 exon) and BnJMJ21;a (9 exon). Subgroup-II keeps highly similar gene structures with 2–3 exons (Fig. 6C). Compared with that in KDM4/JHDM3 group, Phe is replaced by Ala within AtJMJ21 orthology and by Ser within AtJMJ22 orthology in JMJD6 (Fig. 6D).

### Group-KDM3/JHDM2

The KDM3 & JHDM2 group is the largest group with 48 JmjC proteins: 6 from *Arabidopsis* (JMJ24-29), 22 from *B. napus*, 8 from *B. rapa*, 5 from *O. sativa,* and 6 from *B. oleracea* (Fig. 2 and 7A). Group-KDM3 & JHDM2 can be divided into four subgroups: subgroup-I containing 14 *Brassica* members and 3 *Arabidopsis* homologous genes, *AtJMJ25*, *AtJMJ26* and *AtJMJ29*. These proteins have AT-hook motif, RING and DM domains, except JmjC domain. Subgroups-II/III/IV contain AtJMJ27, AtJMJ24 and AtJMJ28 and their homologue genes, respectively. Subgroups-III/IV show highly-conserved and shared JmjC and RING domains, except BoJMJ24;a (Fig. 7B). Moreover, their gene structure also shows corresponding conservation (Fig. 7C).

In group-KDM3/JHDM2 (Fig. 7B), half of the members harbor RING domain as the second primary domain. $Cys-X_2-Cys-X_{(9-39)}-Cys-X_{(1-3)}-His-X_{(2-3)}-Cys-X2-Cys-X_{(4-48)}-Cys-X_2-Cys$ is the canonical RING (*Deshaies & Joazeiro, 2009*). The RING domain of many proteins mainly binds to ubiquitin-conjugating enzymes and mediates the direct transfer of ubiquitin to substrate (*Deshaies & Joazeiro, 2009*). The AT-hook is a small DNA-binding motif with a preference for A/T rich regions found in various proteins, such as the high mobility group proteins (*Klosterman & Hadwiger, 2002*).

Sequence alignment and logos analysis of the JmjC domain reveal that subgroups-I and II are highly conserved in Fe(II) binding sites (His, Asp and Cys) and $\alpha$KG binding sites (Thr and Lys), except BoJMJ29;a. Moreover, both sites of subgroup-IV are different: The His and Asp residues of Fe(II) binding sites are replaced by Gly and Glu residues, and the Thr of $\alpha$KG binding sites is replaced by Lys residue (Fig. 7D). However, subgroup-III does not present conservation.

### Stress-response expression of KDM5 subfamily genes

*Arabidopsis* KDM5 subfamily genes play central roles in stress-responsive gene expression and gene priming by H3K4me3 demethylation (*Jaskiewicz, Conrath & Peterhansel, 2011*). The expression of genes related to the response for drought, high temperature and saline stresses was determined to characterize the corresponding function of KDM5 group homologues in *B. napus* abiotic stress response.

Under three different stress conditions, the expression profiles of BnKDM5 subfamily genes were detected by real-time PCR (Figs. 8–10). *BnJMJ16;a, BnJMJ17;b/c and BnJMJ18;a* showed remarkably elevated expression under salt, drought and high temperature. However, BrJMJ19;a/c did not show significant expression changes. The vast majority of JmjC genes showed remarkably elevated expression under drought treatment, except *BnJMJ14;a* and *BnJMJ19;a/c/e*. Under drought 5 or 10 days, most of the genes had higher expression than drought 15 days, except *BnJMJ17;c* and *BnJMJ19;b* (Fig. 8). However, only 6 (*BnJMJ16;a, BnJMJ17, BnJMJ18;a* and *BnJMJ19;e*) out of the 20 JmjC genes showed elevated expression under high temperature treatment. The expressions of *BnJMJ16;a, BnJMJ17a/b* and *BnJMJ18;a* expression was induced under 12 h of high temperature treatment, but *BnJMJ17;c* and *BnJMJ19;e* were not substantially expressed until 36 h (Fig. 9). Moreover, nearly half of the JmjC genes (*BnJMJ14, BnJMJ15;a, BnJMJ16;a, BnJMJ17;b/c* and *BnJMJ18;a/d*) showed remarkable expression under 100 Mm NaCl treatment, besides BnJMJ15;c that was strongly induced by 200 mM NaCl stress (Fig. 10).

## DISCUSSION

### Conserved JmjC Genes of *B. napus*

Allotetraploid species *B. napus* (AACC, $2n = 38$) derived from interspecific spontaneous hybridization of *B. rapa* (AA, $2n = 20$) and *B. oleracea* (CC, $2n = 18$) (*Nagaharu, 1935*). Nuclear genomes have remained essentially unaltered since amphidiploid species formation (*Parkin et al., 1995*). Similarly, the JmjC protein family appears to be extremely conserved during *B. napus* formation. Compared with the progenitor genomes of *B. rapa* and *B. oleracea*, 27 (93.1%) JmjC orthologous gene pairs between An subgenome and 19 (82.6%)

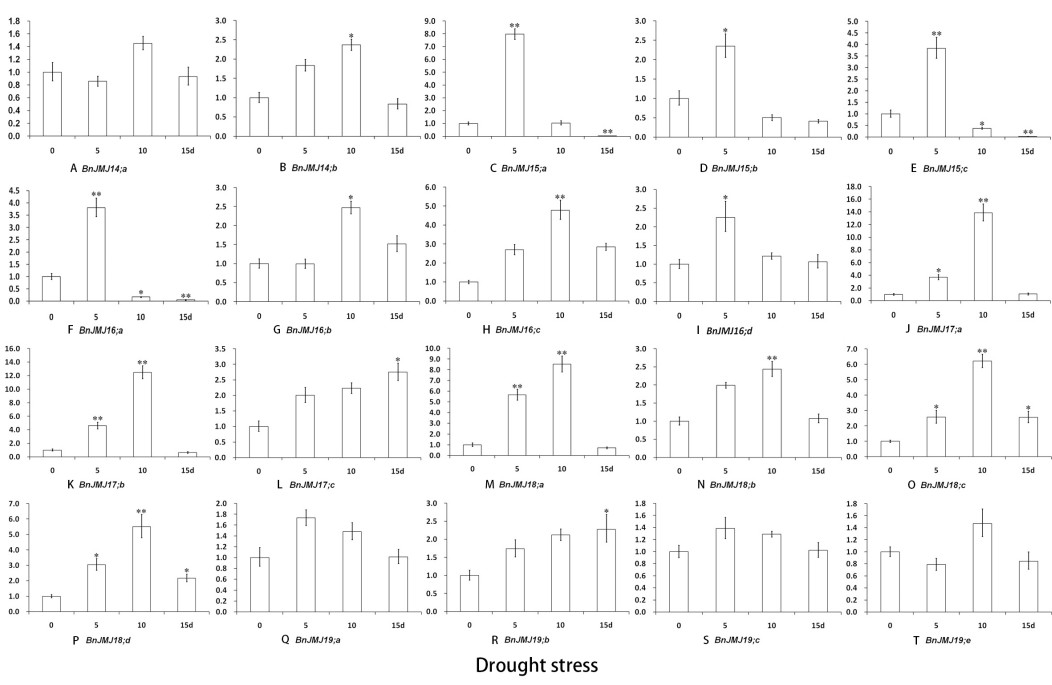

**Figure 8** (A–T) Expression of *B. naups* KDM5 subfamily in response to drought. Many of the BnJMJ14-19 genes involved in drought stress response. The error bars depict SD, an asterisk represent corresponding gene significantly up- or down-regulated by Student's *t* test between the treatment and the control (0.01 < *P* < 0.05), two represent (*p* < 0.01).

between Cn subgenome in *B. napus* were conserved (Fig. 2). The average retention rates from ancestor exceed the rate of all homologous gene pairs (83.7%) across the whole *B. napus* genome (*Chalhoub et al., 2014*). Each member of *B. rapa* and *B. oleracea* can be paired to at least one homologue of *B. napus,* except five members of KDM5A subfamily: *BrJMJ14;b, BrJMJ15;b, BoJMJ15;b, BoJMJ19;c* and *BoJMJ19;d*, which indicates the JmjC genes are highly conserved but some reductions might have been found in KDM5A subfamily during the formation process of allotetraploid (Fig. 4). Comparing with the reported homologous genes, *BrJMJ14;b, BrJMJ15;b* and *BoJMJ15;b* might be associated with floral integrators and flowering time by H3K4 demethylase activities (*Lu et al., 2010*; *Yang et al., 2010*; *Yang et al., 2012b*). However, the *BrJMJ14;b, BrJMJ15;b* and *BoJMJ15;b* might be redundant, because their paralogs can be found in *B. napus* (Fig. 4). In addition, *BoJMJ19;c* and *BoJMJ19;d* have present difference in structures and might only specifically exist in *B. oleracea* (Fig. 4).

Gene duplication expands genome content and changes gene function to ensure the optimal adaptability and evolution of plants (*Xu et al., 2012*). The 65 JmjC proteins from *B. napus* were more than the total number of proteins for *B. rapa* (29) and *B. oleracea* (23) (File S1). According to the systematic analysis results of JmjC proteins, some new or duplicated JmjC genes were found in *B. napus* (Figs. 2–7). Gene duplication events were confirmed by the method of *Yang et al. (2008)* and *Sun et al. (2015)*. *BnJMJ16;e/BnJMJ16;d, BnJMJ18;a/BnJMJ18;b, BnJMJ18;d/BnJMJ18;c,*

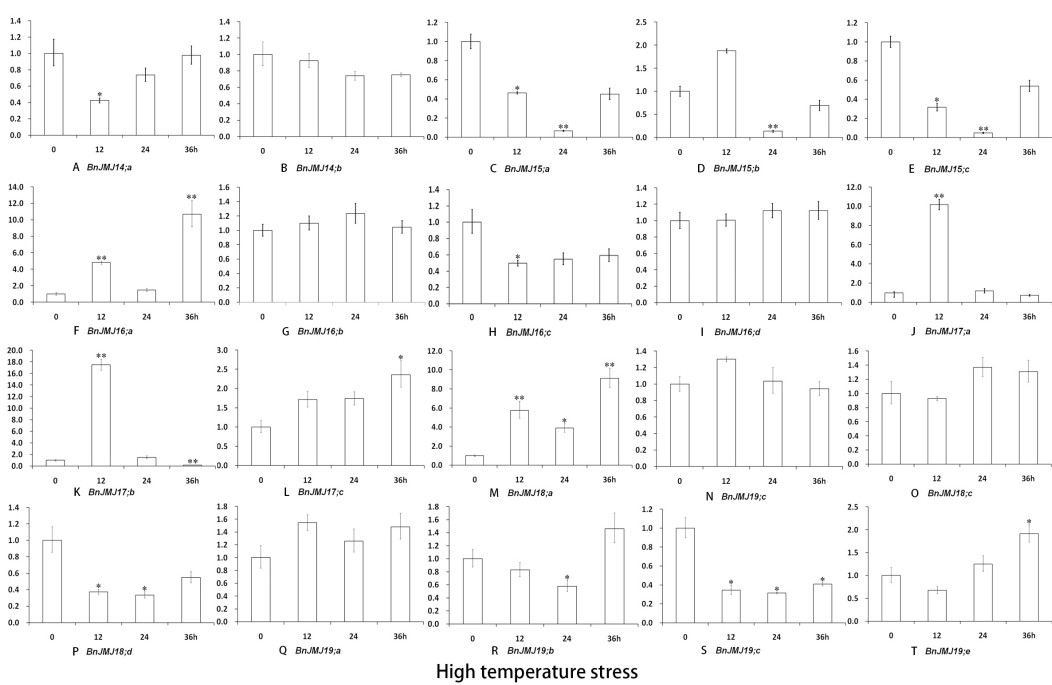

**Figure 9** **(A–T) Expression of *B. naups* KDM5 subfamily in response to high temperature.** Many of the BnJMJ14-19 genes involved in high temperature stress response. The error bars depict SD, an asterisk represent corresponding gene significantly up- or down-regulated by Student's *t* test between the treatment and the control ($0.01 < P < 0.05$), two represent ($p < 0.01$).

*BnJMJ31;a/BnJMJ31;b*, *BnJMJ29;b/BnJMJ29;d* and *BnJMJ17;a/BnJMJ17;b* duplicated genes pairs may have been derived from the existing JmjC gene from *B. rapa* and *BnJMJ28;a/BnJMJ28;b and BnJMJ29;a/BnJMJ29;c* pairs from *B. oleracea* (File S1). These gene pairs were duplicated through segmental duplication (File S2). Additionally, the parent of *BnJMJ17;c* was not found by the method, but its sequence of JmjC domain was consistent with the *BnJMJ17;b*. In crop species, gene duplicate events can contribute to the evolution of novel functions and important agronomic traits, such as fruit shape, flowering time, disease resistance and adaptation to stress (*Panchy, Lehti-Shiu & Shiu, 2016*). In contrast, whole genome triplication event of *Brassica rapa* and *Brassica oleracea* exerts critical roles in the speciation and morphotype diversification of *Brassica* plants (*Cheng, Wu & Wang, 2014*).

Overall, the JmjC genes of *B. napus* were conserved during the formation process of allotetraploidy, and the gene reduction and duplication from parents were preferred in the KMD5A group.

## Conservation and function of JmjC proteins of *B. napus*

65 JmjC proteins of *B. napus* were clustered into seven groups based on phylogenetic and domain organization (Figs. 2–7; File S1) similar to the result that JmjC domain proteins is systematic analyzed ranging from green alga to higher plant (*Huang et al., 2016*). Furthermore, the BnJmjC demonstrated high similarity with homologous sequences or

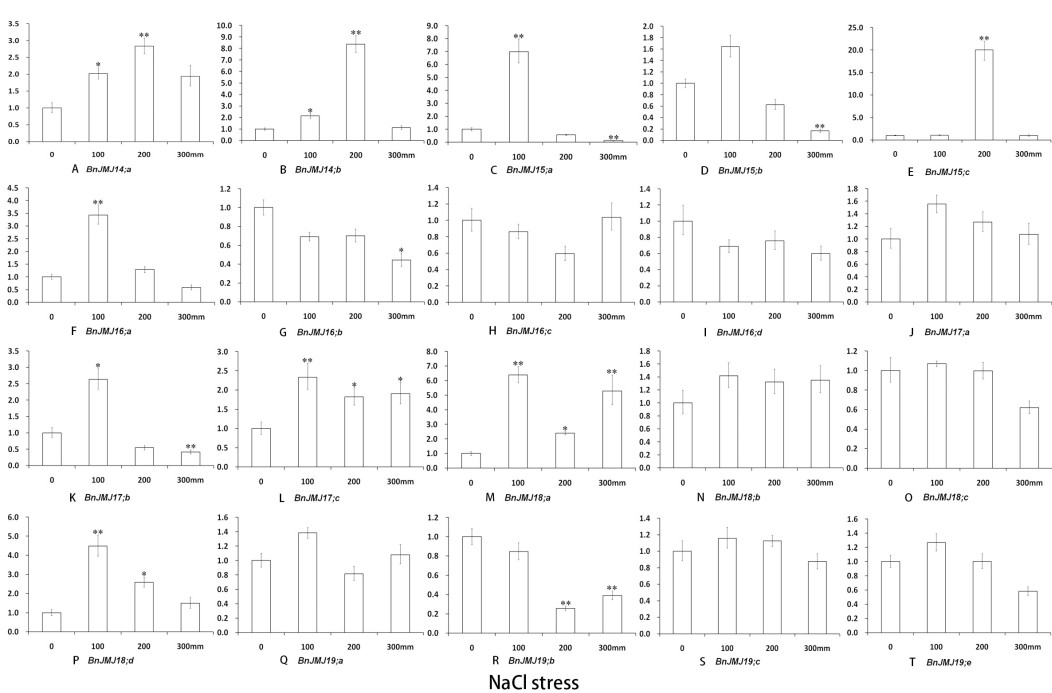

**Figure 10** (A–T) Expression of *B. naups* KDM5 subfamily in response to NaCl stresses. Many of the BnJMJ14-19 genes involved in NaCl stress response. The error bars depict SD, an asterisk represent corresponding gene significantly up- or down-regulated by Student's *t* test between the treatment and the control (0.01 < *P* < 0.05), two represent (*p* < 0.01).

even with the whole subfamily in domain origination, chromosomal location, intron/exon number and catalytic sites. These results indicated JmjC proteins of *B. napus* were conserved family during allotetraploid formation.

In general, H3K4me2/me3 and H3K36 correlate with transcriptional activation, and H3K9me2 and H3K27me3 correlate with gene silencing (*Huang et al., 2011*). The substrate specificity of BnJmjC proteins can be predicted based on their conservation and previous research results. KDM4/JHDM3 was involved in multi-demethylation (H3K4me2/3, H3K9me3, H3K27m2/3 and H3K36me2/3), such as AtJMJ11 for H3K27m3, H3K9me3 and H3K4me3 demethylation (*Jeong et al., 2009*; *Crevillén et al., 2014*; *Noh et al., 2004*; *Yu et al., 2008*), AtJMJ12/REF6 for H3K4me2/3, H3K27me2/3 and H3K36me2/3 demethylation (*Cui et al., 2016*; *Hou et al., 2014*; *Hyun et al., 2016*; *Ko et al., 2010*; *Li et al., 2016*; *Lu et al., 2011*) and OsJM12;a/JMJ705 for H3K27me2/3 demethylation (*Li et al., 2013*). KDM5A was involved in H3K4me2/3 demethylase activity, and the activity has been reported in AtJMJ14/15/18 and JMJ703(OsJM16;a)/ JMJ704/OsJM14;a (*Lu et al., 2010*; *Yang et al., 2010*; *Yang et al., 2012b*; *Shen et al., 2014*; *Yang et al., 2012a*; *Chen et al., 2013*; *Cui et al., 2013*; *Hou et al., 2015*). JmjC domain-only B is involved in H3K36me2 and H3K27me3 demethylation, but only a member (AtJMJ30) is identified in this subfamily (*Yan et al., 2014*; *Gan et al., 2014*). KDM3/JHDM2 is involved in H3K9 demethylation, and the demethylase activity has been reported in AtJMJ25/27 (*Dutta et al., 2017*; *Saze, Sasaki & Kakutani, 2014*). KDM5B might be associated with H3K4 demethylase by BRIGHT and

PHD domains, but there are still no reports on the members of this family. Moreover, JmjC domain-only A subfamily member AtJMJ20 has a crucial role in removing histone arginine methylases (*Cho et al., 2012*). JMJD6 subfamily member AtJMJ22 acts as histone arginine demethylases (*Cho et al., 2012*).

### KDM5 response to abiotic stresses

Epigenetic marks in H3K4 exert critical functions on regulating genes response to ambient stress (*Baulcombe & Dean, 2014*; *Begcy & Dresselhaus, 2018*). In rice, H3K4 dimethylation of *ADH1* and *PDC1* is switched to trimethylation to response to submergence stress (*Qiao & Fan, 2011*). H3K4me3 is also correlated with gene expression which responds to dehydration stress (*Santos et al., 2011*). *Arabidopsis* H3K4me3 of *AHG3*, catalyzed by ATX4 and ATX5, plays an essential role in drought stress response (*Liu et al., 2018a*; *Liu et al., 2018b*). *Arabidopsis* H3K4 hypermethylation is associated with transcriptional activation and maintenance heat stress response (*Liu et al., 2018a*; *Liu et al., 2018b*). Furthermore, Over-expression of KDM5 subfamily AtJMJ15, a H3K4 demethylase, enhanced salt tolerance (*Shen et al., 2014*). In KDM5 subfamily, the similar gene sequences and domain organization between *Arabidopsis* and *B. napus* suggests that *B. napus* members may also possess conserved biological function with H3K4 demethylase activity (Fig. 4). The expression patterns of BnKDM5 subfamily show that almost all of *BnKDM5* genes are involved in drought, high temperature and salt stress response (Figs. 8–10).

Under drought, high temperature or salt stress, all members of *BnKDM5B* exhibited remarkable elevated expression, except *BnJMJ17;a* under salt stress, suggesting that these homologous genes have conserved functions to responses to similar stress stimuli. In general, the expression of *BnKDM5A* genes are relatively conserved to response to identical stress condition. For instance, the homologous gene of *AtJMJ15* (*BnJMJ15;a/b/c*), *AtJMJ16* (*BnJMJ16;a/b/c/d*), and *AtJMJ18* (*BnJMJ18;a/b/c/d*) showed similar stress response to drought stress with remarkable increased expression. However, their display diverse transcriptional responses to other stress stimuli, even among homologous genes. For example, *BnJMJ18;a* shows remarkably elevated expression under high temperature without the homologous genes *BnJMJ18;b/c/d*. These results indicate that functions of *BnKDM5* members are conserved and divergent during allotetraploid formation.

## CONCLUSIONS

This study provides the first genome-wide characterization of JmjC genes in *Brassica* species. The *BnJmjC* exhibits higher conservation during the formation process of allotetraploid than the average retention rates of whole *B. napus* genome. Furthermore, expression profiles indicated that *BnKDM5* subfamily genes are involved in stress response to salt, drought and high temperature.

## ACKNOWLEDGEMENTS

We are grateful to the Brassica database and its contributors who provided their data for this analysis.

### Funding

This work was funded by the National Nature Science Foundation of China (31971834) and the Natural Science Foundation of Hunan Province (2019JJ40116). The funders had no role in study design, data collection and analysis, decision to publish, or preparation of the manuscript.

### Grant Disclosures

The following grant information was disclosed by the authors:
National Nature Science Foundation of China: 31971834.
Natural Science Foundation of Hunan Province: 2019JJ40116.

### Competing Interests

The authors declare there are no competing interests.

### Author Contributions

- Xinghui He conceived and designed the experiments, performed the experiments, analyzed the data, prepared figures and/or tables, authored or reviewed drafts of the paper, and approved the final draft.
- Qianwen Wang, Jiao Pan and Boyu Liu performed the experiments, authored or reviewed drafts of the paper, and approved the final draft.
- Ying Ruan conceived and designed the experiments, authored or reviewed drafts of the paper, and approved the final draft.
- Yong Huang conceived and designed the experiments, authored or reviewed drafts of the paper, and approved the final draft.

### Data Availability

Raw data are available in the Supplementary Files.

### Supplemental Information

Supplemental information for this article can be found online at http://dx.doi.org/10.7717/peerj.11137#supplemental-information.

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
