# Peer review of "Systematic analysis of JmjC gene family and stress­-response expression of KDM5 subfamily genes in Brassica napus"

_PeerJ, doi:10.7717/peerj.11137_

## Round 0.1 · original submission · Major Revisions

We request that you make major revisions based on reviewer's comments and my suggestions before it is processed further.

1 Please reorganize and improve the logic in the Abstract. Introduction of evolution of jmjC proteins in diverse plants is also necessary.

2 The authors should describe meaning, brief summary of methods and results in the Abstract.

3 Identification of JMJC proteins should be reanalyzed based on at least two methods, such as BLASTP and Hmm research. Parameters used in the programs should be described for the repeatability.

4 As an evolutionary research, BI and ML trees should be constructed. Then, the best topology should be selected to determine the phylogenetic relationship among proteins.

5 To ensure the effectiveness of stress treatment, both the figures and measurement of physiological traits, such as enzyme activity should be provided. The authors sampled leaves at different treatment time points, how about untreated controls?

6 Why there isn’t internal control in the qRT-PCR analysis? Please also provide the evidence gene-specificity of qRT-PCR primers.

7 The authors should check the distribution of jmjC genes on the six ancient sub-genomes in B. napus.

8 Gene duplication and whole genome triplication in the evolution jmjC genes of Brassica crops and other plants should be compared in the discussion.

9 In the “Conservation and Function of JmjC Proteins of B. napus” section, there are too many references from line 382 to 421. But you only discussed JmjC Proteins of B. napus in the 11 lines, with one reference. I think this is not true discussion.

Reviewer 1 ·

Basic reporting

This manuscript studied Jumonji C (JmjC) proteins in plant development and stress response
through the removal of lysine methylation from histones. Real-time quantitative polymerase chain reaction (RT-qPCR) and some bioinformatic prediction tools were used to support the functional Group-KDM5 genes responded to in various stress responses.
The manuscript provided basic provide a basis for future functional characterization of the Jumonji C (JmjC) proteins in stress response.

The introduction does well organize paragraphs in order to emphasize the necessity of this study. Literature is well referenced and relevant on studied topic. Discussion part repeated some data mentioned in result part. Shorten discussion part is recommended.

Experimental design

Methods was described with sufficient detail and experimental/bioinformatics tools used in this research was well defined, relevant and meaningful.

Validity of the findings

The impact and novelty of this manuscript are assessed. Overall, this manuscript was qualified for publication with minor revision.
Minor comments:
Figure 8: the graphs are too small to read out. I recommend that the data can be illustrated as different figures for drought, high temperature and NaCl stress. Adding axis title

·

Basic reporting

The authors present a systematic analysis in Brassica napus of the JmjC gene family and an analysis of the KDM5 subfamily gene responses to stress. They made a comparative analysis of the genes members conservation based on phylogeny and protein structure/motif comparison and using the parental genome relatives. Although the bootstrap values were sometime weak, the topology found was coherent with previous work or confirmed by analyses focused on each clade. Therefore, the presented data is sound as an analysis of the redundancy and expansion of the JmJC gene family in the alloaphiployd B. napus and not for an evolutionary work as is stated in some sentences. Such study provides a frame to analyse the functional conservation of a group of paralogs as provided by the authors, by studying the response of the KDM5 members to stress.

Experimental design

The experiments was well conducted

Validity of the findings

The data are sound and well support the manuscript. However, they do not support any analysis of family evolution and should remain as a comparative analysis of the JmJC in Brassica napus. As stated before, the bootstrap values are low and only the large clade and few subclades are support. So all statement about family evolution should be remove from the text, especially when the authors do not discuss the low value of some branches. In addition, a paragraph of the discussion is redundant and should be remove for a clear reading of the manuscript.

Additional comments

The presented data is sound as an analysis of the redundancy and expansion of the JmJC gene family in the alloaphiployd B. napus and not for an evolutionary work as is stated in some sentences. Such study provides a frame to analyse the functional conservation of a group of paralogs as provided by studying the response of the KDM5 members to stress.

Major comments
Figure 8 is of bad quality and could not be read and checked
Rewrite the end of the background in the abstract, please. The sentence starting line 80 is better appropriate “The protein organization and function of JmjC domain in Brassica species and its relative relationship with model plant Arabidopsis….”
Line 185, please rewrite the last sentence of the paragraph
Line 366 please add “might indicate” instead of indicate
It is not clear to me what is discuss in beginning of the paragraph “Conservation and Function of JmjC Proteins of B. napus” starting line 381. The information is repetitive with the above introduction and results. I suggest the author to remove this part until line 421
Instead, the author could discuss about the reduction/expansion of some clade depending of the function of their known paralogues in Arabidopis or Rice development or about similar observation found for the mads-box family by Wu et al (even if their title is not right).
Line 425 should be rewritten and the evolution idea removed
Minor comments
Line 127, add the substrates used
Please indicate the title of the C column in supplementary data1
Sup figure have poor bootstrap value and cannot be used to draw any conclusion
Line 475 replace evolutional by divergent

---

## Round 0.2 · Major Revisions

Please make substantial improvements to the current manuscript as indicated by the reviewer.

·

Basic reporting

no comment

Experimental design

no comment

Validity of the findings

no comment

Additional comments

JmjC homologous proteins play critical roles in plant development and stress response. The manuscript by He et al. reports a genome-wide survey and analyses of this gene family in Brassica napus genome. Firstly, the work performed a genome-wide identification of the JmjC homologous genes in B. napus genome, and then performed the phylogenetic and genomic analysis (chromosomal distribution, gene and protein structure analysis) of the candidates, followed by the expression profile analysis of the KDM5 subfamily under drought, high temperature, and NaCl stresses by qRT-PCR. The methods in this study are relevant and are well applied for most of the study. This type of study falls within the scope of PeerJ. The major results of this study are based on Bioinformatics analysis, which is an important fundament for further functional researches of the candidate BnJmjCs in the future. However, the manuscript is mainly descriptive, and is not well organized with many grammatical in the whole text. Some major points listed below should be corrected before it could be accepted for publication in this journal.

Major comments:
1. Line 1, the title of this ms needs to be correct, as its meaning is obscure.
2. Line 25, “JmjC domain” should be “JmjC proteins”?
3. Line 26, “Arabidopsis” should be in italic. And this problem should be amended in the whole text.
4. Line 28-30, the current content in this part did not describe the main ideas and methods applied in this study well. Thus, it should be re-wrote.
5. “Results” part in “Abstract”, the main results of this study are not summarized well and are not highlighted enough.
6. Line 32, the statement of “These genes were grouped into seven clades base on the analyzation for JmjC sequences” is unconvincing, as mentioned in the main text, the classification of this gene family was based on the phylogenetic analysis?
7. Line 36-37, the statement of “Furthermore, BrKDM5 genes were examined under stress conditions due to their crucial functions in H3K4 demethylation” is meaningless. The authors should focus on the result of this analyses here, and should illustrate the detail stress treatments as well.
8. Overall, the contents in “INTRODUCTION” section is not well organized, especially these in line 59-85. Moreover, the contents in this section is too long, it is better to compress it. The authors should focus on the theme of this study, and highlight the KDMs instead of the other histone modification events.
9. Line 130-133, the description about the identification of candidate JmjC protein sequences is not clear. Moreover, “O. sativa” in line 133 should be in full name instead of abbreviation for the first time in the ms.
10. Line 160, “3 days” should be “at 3 days”?
11. Line 161, “in 40 °C” should be “at 40 °C”?
12. Line 168-169, it is better to use at least two reference genes in the qRT-PCR analysis.
13. Line 177, “C-genomics” and “A-genomics” should be “C subgenome” and “A subgenome” or “Cn subgenome” and “An subgenome”.
14. In the section of “Chromosome Maps of JmjC Genes in Brassica”, it’s better to describe the nomenclature method of the candidates and supplied a table to show the major features of the candidates.
15. Line 196-197, why the proteins should be renamed in this part?
16. Line 197, “hasthe” should be “has the”. Moreover, why the NJ tree is better than the ML tree? What is the reason? In general, it is more credible if the results got from different methods were the same or very similar. Furthermore, in line 139-145, “Analysis of JmjC sequences” in the method section, the authors just described the method used for constructing the NJ tree. The method of constructing ML phylogenetic tree was not supplied.
17. The authors stated in line 101-103 that the Arabidopsis JmjC proteins were divided into five subfamilies. However, in line 199-200, this gene family was divided into seven subfamilies. Why the KDM5 and Jmic domain-only subfamilies were classified into two subfamilies respectively? As shown in Figure 2, the bootstrap value of KDM5A clade is very low, so, what is your criterion to divide the KDM5 subfamily into KDM5A and B subfamilies?
18. Line 244-248, the authors should focus on their own results in the “Result” section. In general, it is not allowed to cite the results from any reference in this part. If there is any extensional meaning about your result, it is better to discuss it in the “Discussion” section.
19. Line 354, the title of “KDM5 Expression in Abiotic Stress” is very strange. What’s your mean? Moreover, there are many grammar mistakes in the contents of this section.
20. Line 355, “KDM5 genes” should be “KDM5 homologous genes”. Accordingly, the authors should uniform the name about this issue in the whole text.
21. Line 364, the meaning of “Under drought 5 or/and 10 day” and “drought 15” are confusing.
22. Line 379, “an subgenome” should be “An subgenome”? And the naming style of the two subgenomes of B. napus should be uniformed in the whole text.
23. Line 380, the authors should describe the method to calculate the “retention rates” in “Method”.
24. Line 396-397, what’s your mean about “Gene duplication events were confirmed by Yang et al. (Yang et al., 2008) and Sun et al. (Sun et al., 2015)”?
25. Line 422, “base on” should be “based on”?
26. Figure 5 is too large. And the authors should indicate the meanings of the labels and/or marks in the figures, such as the triangle in Figure 5H, etc.
27. The letters of A-T are not necessary in Figure 8-10. And the words of “Drought stress”, “High temperature stress” and “ NaCl stress” should be removed from these figures respectively.
28. There are many grammatical mistakes in this paper. I recommend language editing for this paper.

---

## Round 0.3 · accepted · Accept

The authors have addressed all the concerns from reviewers.

·

Basic reporting

no comment

Experimental design

no comment

Validity of the findings

no comment

Additional comments

Comments to the Author (s):

Many of my comments have been addressed. The quality of this manuscript had been greatly improved this time. I think that this new revision can be accepted for publication now. However, there still are some problems which should be addressed before it can be published.

Major comments:
1. As I have stated in the first round, there are many mistakes in English style and grammar that can be found throughout the manuscript. For example, line 30, “synonymous” should be “Synonymous”; line 32, “the expression level” should be “the expression levels”, line 40 “treatment” should be “treatments”; line 345, “day” should be “days”; line 348, “BnJMJ16;a, BnJMJ17a/b and BnJMJ18;a expression was induced” should be “ the expressions of BnJMJ16;a, BnJMJ17a/b and BnJMJ18;a expression were induced…”, etc. Therefore, I suggested the writing needs to be thoroughly checked and substantially improved by an English language editing company, again.

2. line 37, the statement of “exceeded whole genome level” is obscure. It is better to state or define it more clearly to avoid confuse.

3. Line 39-40, the sentence of “Furthermore, BrKDM5 subfamily genes were examined under stress conditions” should be deleted from this section.

4. Line 44, “expression profiles indicated that” could be “expression profiles of many …”

5. Last time, I have suggested that “Overall, the contents in INTRODUCTION section is not well organized. Moreover, the contents in this section is too long, it is better to compress it. The authors should focus on the theme of this study, and highlight the KDMs instead of the other histone modification events.” Although the authors have did some amends in this part, the problems that I mentioned last time are still existed here. The length is too long, such as the second paragraph has up to 48 lines. The contents in this part should focus on “JmjC proteins” or “KDM proteins”. In addition, as we know, the last paragraph in “INTRODUCTION” section is generally about the aim and main researches in the ms. However, these contents were missed in this ms.

6. Line 114, “Oryza sativa sequences” should be “The JmjC protein sequences in Oryza sativa…”.

7. Line 126, “The resulting files” should be “The multiple sequence alignment result…”. Moreover, in general, the bootstrap test in ML analyses is 100 replicates. Are you sure you used “1000 replicates”?

8. Line 128, please check whether you used the CDS sequences of the candidates to create the logo maps? It should be proteins sequences in Figure 3 to 7?

9. Line 161-162, the statement of “and 8 genes were still on scaffolds, in which 5 were from Cn subgenome and 2 from An subgenome” is right? 5+2=7?

10. Line 167-168, “Four tandem JmjC genes pairs located on chromosomes A03, A09, and C03 in B. rapa (Fig. 1).” should be “Four tandem JmjC genes pairs located on chromosomes A03, A09, and C03 in B. napus”?

11. There are still many mistakes in the section of “Stress-response expression of KDM5 subfamily genes”, such as line 324, “under salt, drought and high temperature” should be “under salt, drought and high temperature treatments”; line 345, the meaning of “drought 15” is confusing, etc.

12. The A-D parts in Figure 1 should be presented in four parts instead of A/B and C/D (two parts).